# Early absolute lymphocyte count was associated with one-year mortality in critically ill surgical patients: A propensity score-matching and weighting study

**Duc Trieu Ho**[1]☯, **The Thach Pham**[1]☯, **Li-Ting Wong**[2], **Chieh-Liang Wu**[3,4], **Ming-Cheng Chan**[3,4], **Wen-Cheng Chao**[3,4,5]*

1 Center for Critical Care Medicine, Bach Mai Hospital, Hanoi, Vietnam, 2 Department of Medical Research, Taichung Veterans General Hospital, Taichung, Taiwan, 3 Department of Critical Care Medicine, Taichung Veterans General Hospital, Taichung, Taiwan, 4 Department of post-Baccalaureate Medicine, College of Medicine, National Chung Hsing University, Taichung, Taiwan, 5 Big Data Center, Chung Hsing University, Taichung, Taiwan

☯ These authors contributed equally to this work.

* cwc081@hotmail.com

**Data Availability Statement:** All relevant data are within the manuscript and its Supporting Information files.

## Abstract

### Background

Absolute lymphocyte count (ALC) is a crucial indicator of immunity in critical illness, but studies focusing on long-term outcomes in critically ill patients, particularly surgical patients, are still lacking. We sought to explore the association between week-one ALC and long-term mortality in critically ill surgical patients.

### Methods

We used the 2015–2020 critical care database of Taichung Veterans General Hospital (TCVGH), a referral hospital in central Taiwan, and the primary outcome was one-year all-cause mortality. We assessed the association between ALC and long-term mortality by measuring hazard ratios (HRs) with 95% confidence intervals (CIs). Furthermore, we used propensity score-matching and -weighting analyses, consisting of propensity score matching (PSM), inverse probability of treatment weighting (IPTW), and covariate balancing propensity score (CBPS), to validate the association.

### Results

A total of 8052 patients were enrolled, with their one-year mortality being 24.2%. Cox regression showed that low ALC was independently associated with mortality (adjHR 1.140, 95% CI 1.091–1.192). Moreover, this association tended to be stronger among younger patients, patients with fewer comorbidities and lower severity. The association between low ALC and mortality in original, PSM, IPTW, and CBPS populations were 1.497 (95% CI 1.320–1.697), 1.391 (95% CI 1.169–1.654), 1.512 (95% CI 1.310–1.744), and 1.511 (95% CI 1.310–

**Funding:** This study was supported by Taichung Veterans General Hospitals (TCVGH-112G213, TCVGH-1124401C, 1124403B, and TCVGH-1124401D) and Taiwan National Science and Technology Council (NSTC 112-2314-B-075A-001-MY2 and NSTC 112-2321-B-075A-001). The funders had no role in the study design, data collection and analysis, decision to publish, or preparation of the manuscript.

**Competing interests:** The authors declare no competing interest.

1.744), respectively. Additionally, the association appears to be consistent, using distinct cutoff levels to define the low ALC.

## Conclusions

We identified that early low ALC was associated with increased one-year mortality in critically ill surgical patients, and prospective studies are warranted to confirm the finding.

## Background

Lymphocyte is an essential cell in immunity and leads to various pathophysiological responses in critical illness [1,2]. A low absolute lymphocyte count (ALC) is one of the immune-compromised biomarkers in critically ill patients and has been implicated with vulnerability for secondary infection or infection resulting from opportunistic pathogens in patients who were admitted to the intensive care unit (ICU) [3–6]. Increasing studies have shown the association between ALC and outcomes in critically ill patients, but the majority of studies were conducted in medical or mixed ICUs [7–11]. Few studies have explored the association between low ALC and outcomes in critically ill surgical patients [12–14].

Notably, the aforementioned low ALC-associated immunosuppressed state in acute illness may last after the stabilization of acute illness and contribute to prolonged pathophysiological abnormalities among patients discharged from the ICUs, which have been known as post-intensive care syndrome (PICS) [15,16]. Additionally, significant progress in critical care medicine over the past twenty years has not only decreased mortality rates in ICUs but also resulted in a steadily growing number of patients being discharged from these units [17]. Increasing studies, including our previous studies, have addressed the long-term outcomes of critically ill patients in medical ICUs [18–20]. However, the role of early low ALC in the long-term outcome among critically ill patients, particularly those who were admitted to surgical ICUs, remains unclear.

This study aims to investigate the association between week-one absolute lymphocyte count and one-year mortality in critically ill surgical patients. We linked the 2015–2020 critical care database of Taichung Veterans General Hospital (TCVGH) with the nationwide death-registry profile of Taiwan and then used both Cox regression and propensity score-based analyses to estimate the association between ALC and long-term mortality in critically ill surgical patients.

## Methods

### Ethical approval

The study was performed in accordance with the Declaration of Helsinki [21]. The Institutional Review Board of the TCVGH approved the study and waived the informed consent because all of the data were de-identified (Approval No. CE23045C).

### Patient population and exposure

The data were assessed for research purposes on March 1, 2023 after the approval by the aforementioned IRB. This retrospective cohort study was conducted at TCVGH, a referral hospital with three surgical intensive care units (SICUs) in central Taiwan, and consecutive critically ill patients admitted to the SICUs between 2015 and 2020 were enrolled. Given that we focused on the long-term mortality impact of week-one absolute lymphocyte count in critically ill

surgical patients, we excluded those admitted to ICU for less than 24 hours and organ transplant recipients (Fig 1). The primary exposure in this study is the week-one absolute lymphocyte count. We used the average level of absolute lymphocyte count in patients with more than two sets of ALC within the first week.

## Primary outcome

The primary outcome of interest was the time to all-cause mortality. In the present study, we used the nationwide death registration profile of Taiwan's National Health Insurance Database (NHID) to determine the date of death among the enrolled critically ill surgical patients. The date of death should be highly accurate because Taiwan has mandated compulsory National Health Insurance (NHI) since 1995, and the coverage has been high in the past decade [22]. The censored date was defined as the date of death in the non-survivor group and one year after being admitted to the ICUs in the survivor group.

## Covariates

We used the critical care database of TCVGH to obtain relevant covariates, including demographics, Charlson comorbidity index (CCI), acute physiology and chronic health evaluation (APACHE) II score, presence of shock defined by the usage of vasopressor for longer than

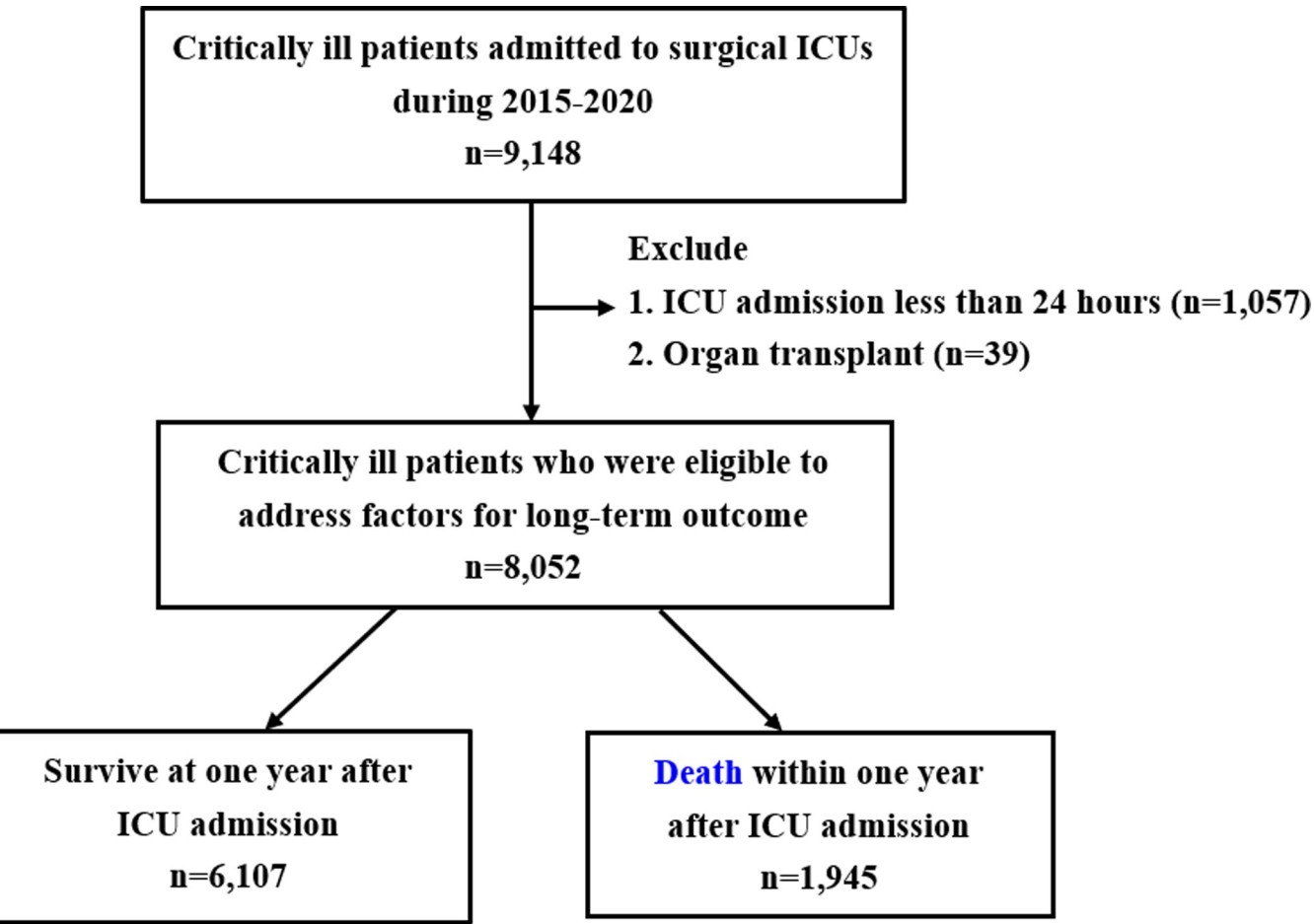

**Fig 1. Flowchart of subject enrollment.** Abbreviations: ICUs, intensive care units; TCVGH.

eight hours, receiving mechanical ventilation for more than three days, receiving renal replacement therapy, laboratory data, surgical divisions, and types of surgery during the admission.

## Statistical analyses

We presented categorical data as numbers (percentages) and continuous data as mean values with standard deviation. Kaplan–Meier plot was used to illustrate the survival time in different subgroups categorised by ALC. Variables were included in the multivariable model if the associated univariable p value was < 0.20 and the variance inflation factor was < 10 [23]. We further included white blood cell count, given that white blood cell count appears to be a known prognosis-relevant variable in critically ill patients [24]. The Cox proportional hazard model was applied to determine the hazard ratios (HRs) for mortality and 95% confidence interval (CIs) after adjusting for potential confounders such as age, sex, comorbidities, surgical divisions, types of surgery, and other potential confounders. We further employed not only propensity score-matching (PSM) but also propensity score-weighting methods, including the inverse probability of treatment weighting (IPTW) and the covariate balancing propensity score (CBPS), to confirm the relationship between the low ALC and high one-year mortality among the enrolled critically ill surgical patients [25–27]. In PSM, we used the optimal nearest neighbour matching algorithm with a calliper distance of 0.15 for the standard mean difference. We used R (version 3.6.0) to analyse the data with the significance level set at 0.05.

## Subgroup analysis

We deployed the Wald test to measure the significance of the modification effect of variables and to determine whether the association between low ALC and high one-year mortality might differ across the variables in this study.

## Sensitivity analyses

The sensitivity analysis was conducted to test the consistency of the association between one-year mortality and the low ALC using distinct cutoff values to define the low ALC in both original and propensity score-matched populations.

## Results

### Characteristics of the enrolled patients

A total of 8502 patients were eligible for analyses in this study (Fig 1). The patient's average age was 61.0±16.0 years, and 62.6% of them were male (Table 1 and S1 Dataset). The proportion of surgical divisions of neurosurgical division, cardiovascular surgical division, divisions for major abdomen surgery consisting of general surgery as well as colorectal surgical divisions and the other surgical divisions were 49.8%, 19.3%, 13.1%, and 17.8%, respectively. We found that 72.6% of patients underwent surgery during ICU admission, with 18.3% of them were emergent surgery. The hospital mortality, 90-day mortality, and one-year mortality in this study were 11.4%, 14.8%, and 24.2%, respectively, and the one-year post-hospital survival rate was hence 14.4% (1028/7135). Compared with the survivor group, patients deceased within one year were older (66.4±15.3 vs 59.2±15.9 years, p < 0.001), had higher CCI (2.3±1.5 vs 1.5 ±1.4, p < 0.001), were more likely to receive major abdominal surgery (24.2% vs 9.6%, p<0.001), and were less likely to receive scheduled surgery (41.4% vs 58.4%, p<0.001). Patients deceased within one year had a higher APACHE II score (23.1±7.0 vs 19.4± 6.7, p < 0.001) and were more likely to have shock (19.7% vs 8.4%, p<0.001), to receive mechanical ventilation (63.2% vs 46.7%, p<0.001) and to receive the renal replacement therapy (16.2% vs 3.3%,

**Table 1. Characteristics of the enrolled critically ill surgical patients divided by the one-year survival.**

| | All | Survivors | Non-survivor | p-value |
|---|---|---|---|---|
| | (N = 8,052) | (n = 6,107) | (n = 1,945) | |
| **Demographic and comorbidity** | | | | |
| Age, years | 61.0 ± 16.0 | 59.2 ± 15.9 | 66.4 ± 15.3 | <0.001 |
| Sex (male), number (%) | 5041 (62.6%) | 3723 (61%) | 1318 (67.8%) | <0.001 |
| Body mass index (kg/m2) | 24.2 ± 4.5 | 24.5 ± 4.5 | 23.3 ± 4.6 | <0.001 |
| Charlson Comorbidity Index (CCI) | 1.6 ± 1.4 | 1.5 ± 1.4 | 2.3 ± 1.5 | <0.001 |
| **Surgical divisions** | | | | |
| Neurosurgical division | 4012 (49.8%) | 3289 (53.9%) | 723 (37.2%) | <0.001 |
| Cardiovascular surgical division | 1553 (19.3%) | 1360 (22.3%) | 193 (9.9%) | <0.001 |
| General-colorectal surgery divisions | 1057 (13.1%) | 586 (9.6%) | 471 (24.2%) | <0.001 |
| Other surgical divisions | 1430 (17.8%) | 872 (14.3%) | 558 (28.7%) | <0.001 |
| **Types of surgery** | | | | |
| Emergent surgery | 1472 (18.3%) | 1103 (18.1%) | 369 (19%) | 0.384 |
| Scheduled surgery | 4373 (54.3%) | 3567 (58.4%) | 806 (41.4%) | <0.001 |
| **Severity and managements** | | | | |
| APACHE II score | 20.2 ± 6.9 | 19.4 ± 6.7 | 23.1 ± 7.0 | <0.001 |
| Presence of shock | 898 (11.2%) | 515 (8.4%) | 383 (19.7%) | <0.001 |
| Receiving mechanical ventilation | 4079 (50.7%) | 2849 (46.7%) | 1230 (63.2%) | <0.001 |
| Receiving RRT | 519 (6.4%) | 203 (3.3%) | 316 (16.2%) | <0.001 |
| End-stage renal disease | 112 (1.4%) | 68 (1.1%) | 44 (2.3%) | <0.001 |
| **Laboratory data** | | | | |
| White blood cell count ($10^3$/μL) | 14.5 ± 3.5 | 14.5 ± 3.4 | 14.5 ± 3.8 | 0.623 |
| Absolute lymphocyte count ($10^3$/μL) | 2.4 ± 1.2 | 2.5 ± 1.2 | 2.0 ± 1.1 | <0.001 |
| Haemoglobin (g/dL) | 13.5 ± 1.6 | 13.6 ± 1.6 | 13.1 ± 1.5 | <0.001 |
| Platelet ($10^3$/μL) | 290.4 ± 58.5 | 291.0 ± 58.4 | 288.5 ± 58.7 | 0.093 |
| Albumin (g/dL) | 3.7 ± 0.5 | 3.7 ± 0.5 | 3.5 ± 0.5 | <0.001 |
| Creatinine (mg/dL) | 1.1 ± 0.4 | 1.1 ± 0.4 | 1.2 ± 0.4 | <0.001 |
| **Outcomes** | | | | |
| ICU length of stay, days | 8.9 ± 8.0 | 7.9 ± 7.4 | 11.9 ± 9.0 | <0.001 |
| Hospital length of stay, days | 18.7 ± 9.1 | 18 ± 8.8 | 20.9 ± 9.6 | <0.001 |
| Hospital mortality | 917 (11.4%) | NA | 917 (47.1%) | NA |
| 90-day mortality | 1193 (14.8%) | NA | 1193 (61.3%) | NA |
| One-year mortality | 1945 (24.2%) | NA | 1945 (100%) | NA |

Abbreviations: CCI, Charlson comorbidity index; APACHE, acute physiology and chronic health evaluation; RRT, renal replacement therapy; ICU, intensive care unit; NA, not available.

p<0.001) than those in the survivor group. Regarding the laboratory data, the level of haemoglobin (13.1±1.5 vs 13.6±1.6 g/dL, p<0.001) and absolute lymphocyte count (2.0±1.1 vs 2.5 ±1.2 $10^3$/μL, p<0.001) in patients deceased within one year was lower than those in the survivor group. In contrast, the white blood cell count and platelet count were similar between the two groups.

## Mortality association of ALC and the subgroup analysis for modification effect

We used the Cox proportional hazard regression to clarify the independent association between the low ALC and long-term mortality among the 8,052 critically ill surgical patients.

We found that a low ALC was independently associated with increased one-year mortality (adjHR 1.140, 95%CI 1.091–1.192, p<0.001) after adjusting for covariates (Table 2). We further evaluated modification effects of covariates and found that the strength of association between ALC and one-year mortality appeared stronger among patients with younger age, higher body mass index, fewer comorbidities, and lower severity of critical illness, including without the need for renal replacement therapy, higher levels of haemoglobin, and lower level of serum creatinine (Table 3).

## Propensity score-matching and weighting analyses

To employ the propensity score-based analyses, we had to divide the patients into high and low ALC groups. Previous studies have used the lymphocyte count lower than 1000/μL to define lymphopenia [2,13,14]; therefore, we chose the cutoff value of low ALC as <1,000 counts/μL in the following propensity score-based analyses. A total of 1296 patients were eligible for the PSM analysis after matching demographics, comorbidities, surgical divisions, types of surgery, severities of critical illness, and laboratory data (Fig 2). The standardised mean difference (SMD) measurement of variables between the two groups demonstrated the high matching quality in this study (S1 Table and S1 Fig). The Kaplan–Meier plots illustrate the significant mortality difference between patients with high and low week-one ALC in both original population and propensity score-matched population (S2 Fig). Table 4 revealed the consistent strength of the association between low ALC and the risk for long-term mortality in

**Table 2. Cox proportional hazard regression analysis for long-term overall mortality among the 8,052 critically ill surgical patients.**

| | Univariable Analysis | | Multivariable Analysis | |
|---|---|---|---|---|
| | HR (95% CI) | p-value | HR (95% CI) | p-value |
| **Demographic and comorbidity** | | | | |
| Age, years | 1.027 (1.024–1.030) | <0.001 | 1.009 (1.006–1.012) | <0.001 |
| Sex (male) | 1.289 (1.172–1.418) | <0.001 | 1.160 (1.050–1.282) | 0.004 |
| Body mass index, (per 1 kg/m2 increment) | 0.949 (0.938–0.959) | <0.001 | 0.964 (0.954–0.974) | <0.001 |
| CCI ≥ 2 | 2.665 (2.427–2.927) | <0.001 | 1.886 (1.710–2.081) | <0.001 |
| **Surgical divisions** | | | | |
| Cardiovascular surgical division | Reference | | Reference | |
| Neurosurgical division | 1.500 (1.280–1.758) | <0.001 | 1.125 (0.946–1.337) | 0.182 |
| General-colorectal surgery divisions | 4.376 (3.700–5.174) | <0.001 | 1.897 (1.590–2.263) | <0.001 |
| **Scheduled surgery** | 0.520 (0.475–0.569) | <0.001 | 0.705 (0.639–0.778) | <0.001 |
| **Severity and managements** | | | | |
| APACHE II score | 1.075 (1.068–1.081) | <0.001 | 1.012 (1.005–1.020) | 0.002 |
| Presence of shock | 2.358 (2.108–2.637) | <0.001 | 0.907 (0.809–1.018) | 0.098 |
| Receiving mechanical ventilation | 1.612 (1.451–1.792) | <0.001 | 1.134 (1.008–1.276) | 0.037 |
| Receiving RRT | 4.144 (3.672–4.677) | <0.001 | 1.763 (1.543–2.013) | <0.001 |
| **Laboratory data** | | | | |
| WBC, per $10^3$/μL increment | 1.006 (0.993–1.019) | 0.399 | 1.008 (0.995–1.021) | 0.216 |
| Haemoglobin (g/dL), per 1 increment | 0.812 (0.787–0.837) | <0.001 | 0.937 (0.908–0.967) | <0.001 |
| Albumin (g/dL), per 1 increment | 0.504 (0.462–0.550) | <0.001 | 0.797 (0.721–0.867) | <0.001 |
| Creatinine (mg/dL), per 1 increment | 2.294 (2.064–2.551) | <0.001 | 1.258 (1.120–1.413) | <0.001 |
| ALC, per $10^3$ /μL decrement | 1.442 (1.379–1.507) | <0.001 | 1.140 (1.091–1.192) | <0.001 |

Abbreviations: CCI, Charlson comorbidity index; APACHE, acute physiology and chronic health evaluation; RRT, renal replacement therapy; WBC, white blood cell count; ALC, absolute lymphocyte count.

**Table 3.  Stratified analyses of modification effect on the association between the absolute lymphocyte count cut by 1000 count/μL and risk of mortality.**

| Variables | Crude HR (95% CI) | p value | Adjusted HR (95% CI) | p value |
|---|---|---|---|---|
| **Age, years** | | <0.001 | | <0.001 |
| <65 | 3.520 (2.966–4.178) | | 2.964 (2.494–3.523) | |
| ≥65 | 1.898 (1.608–2.241) | | 1.817 (1.539–2.145) | |
| **Sex** | | 0.530 | | 0.763 |
| Female | 2.729 (2.197–3.389) | | 2.243 (1.804–2.788) | |
| Male | 2.524 (2.189–2.911) | | 2.272 (1.969–2.622) | |
| **Body mass index**, (kg/m2) | | 0.003 | | 0.005 |
| <23.8 | 2.216 (1.906–2.576) | | 1.941 (1.668–2.258) | |
| ≥23.8 | 3.118 (2.564–3.792) | | 2.664 (2.188–3.243) | |
| **CCI ≥ 2** | | 0.004 | | 0.013 |
| <2 | 3.009 (2.439–3.712) | | 2.671 (2.161–3.300) | |
| ≥2 | 2.093 (1.811–2.419) | | 2.082 (1.802–2.407) | |
| **Surgical divisions** | | 0.798 | | 0.998 |
| Neurosurgical division | 2.384 (1.566–3.630) | | 1.878 (1.230–2.867) | |
| Cardiovascular surgical division | 2.118 (1.683–2.665) | | 1.967 (1.562–2.476) | |
| General-colorectal surgery divisions | 1.891 (1.495–2.393) | | 1.812 (1.431–2.295) | |
| **APACHE II score** | | 0.292 | | 0.317 |
| <20.0 | 2.797 (2.224–3.518) | | 2.383 (1.893–2.999) | |
| ≥20.0 | 2.366 (2.058–2.720) | | 2.105 (1.830–2.422) | |
| **Presence of shock** | | 0.683 | | 0.966 |
| No | 2.497 (2.174–2.868) | | 2.152 (1.873–2.473) | |
| Yes | 2.276 (1.793–2.887) | | 2.117 (1.667–2.687) | |
| **Receiving mechanical ventilation** | | 0.361 | | 0.390 |
| No | 2.899 (2.249–3.736) | | 2.443 (1.893–3.151) | |
| Yes | 2.471 (2.159–2.828) | | 2.156 (1.883–2.469) | |
| **Receiving RRT** | | <0.001 | | <0.001 |
| No | 2.721 (2.384–3.105) | | 2.407 (2.109–2.749) | |
| Yes | 1.130 (0.857–1.489) | | 1.126 (0.853–1.486) | |
| **White blood cell count ($10^3$ /μL)** | | 0.551 | | 0.527 |
| <14.7 | 2.800 (2.404–3.261) | | 2.446 (2.098–2.851) | |
| ≥14.7 | 2.544 (2.089–3.096) | | 2.224 (1.826–2.709) | |
| **Hemoglobin (g/dL)** | | 0.009 | | 0.008 |
| <13.3 | 2.184 (1.887–2.529) | | 1.915 (1.653–2.218) | |
| ≥13.3 | 3.022 (2.459–3.714) | | 2.645 (2.150–3.253) | |
| **Platelet ($10^3$/μL)** | | 0.247 | | 0.756 |
| <287.0 | 2.435 (2.069–2.865) | | 2.223 (1.889–2.617) | |
| ≥287.0 | 2.830 (2.376–3.371) | | 2.322 (1.947–2.771) | |
| **Albumin (g/dl)** | | 0.967 | | 0.995 |
| <3.6 | 2.436 (2.098–2.828) | | 2.160 (1.858–2.509) | |
| ≥3.6 | 2.481 (2.034–3.028) | | 2.126 (1.741–2.596) | |
| **Creatinine (mg/dL)** | | <0.001 | | <0.001 |
| <1.5 | 2.914 (2.529–3.356) | | 2.525 (2.191–2.911) | |
| ≥1.5 | 1.657 (1.329–2.067) | | 1.593 (1.277–1.988) | |

Abbreviations: HR, hazard ratio; CI, confidence interval; CCI, Charlson comorbidity index; APACHE, acute physiology and chronic health evaluation; RRT, renal replacement therapy.

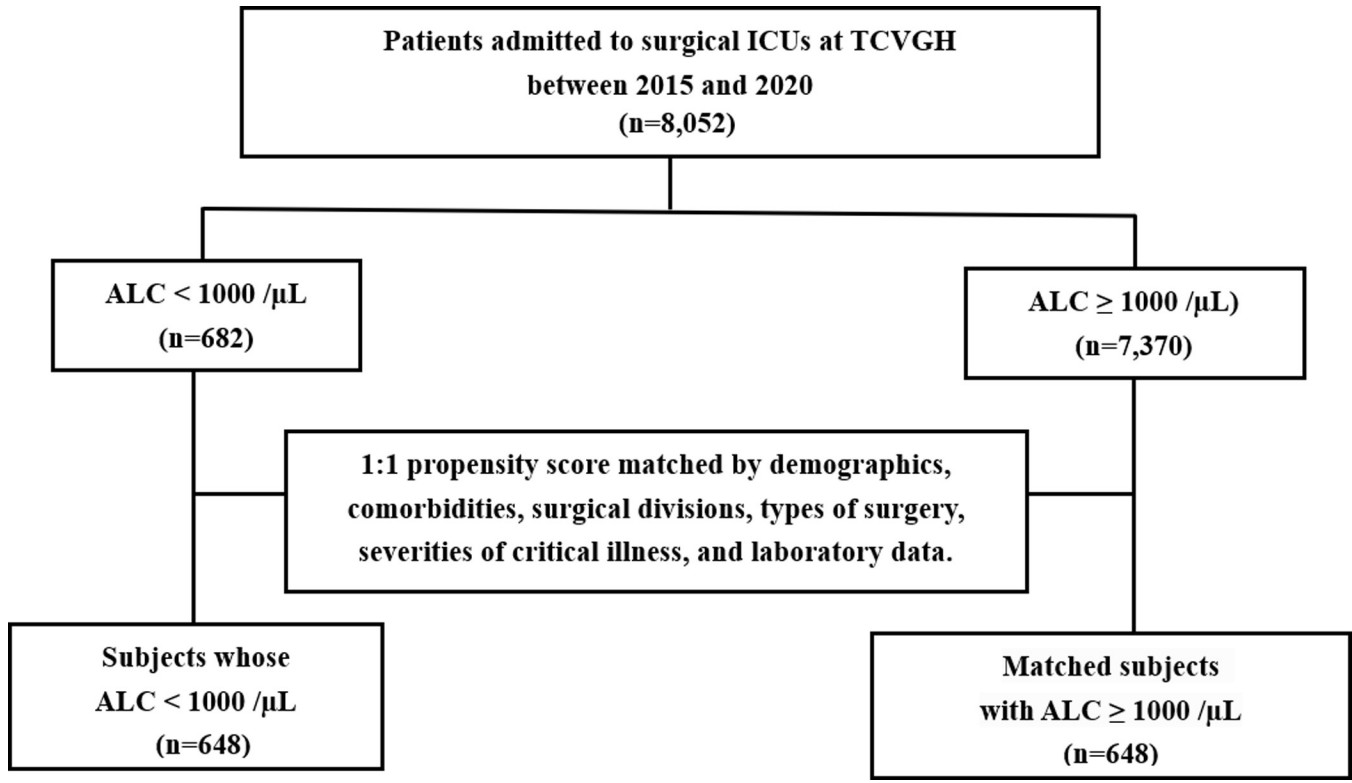

**Fig 2. Flowchart of propensity score matching.** Abbreviations: ALC, absolute lymphocyte count; ICUs, intensive care units; TCVGH, Taichung Veterans General Hospital.

the four populations. The adjusted HRs in original, PSM, IPTW, and CBPS populations were 1.497 (95% CI 1.320–1.697, p<0.001), 1.391 (95% CI 1.169–1.654, p<0.001), 1.512 (95% CI 1.310–1.744, p<0.001), 1.511 (95% CI 1.310–1.744, p<0.001), respectively.

**Table 4. Cox proportional hazard regressions for estimation of the association between level of absolute lymphocyte count cut by 1000 counts/μL and long-term all-cause mortality in distinct patient populations.**

|  | Primary population | | PSM | | IPTW cohort | | CBPS cohort | |
| --- | --- | --- | --- | --- | --- | --- | --- | --- |
|  | HR (95%CI) | | HR (95%CI) | | HR (95%CI) | | HR (95%CI) | |
| **Mode 1** | 2.611 (2.318–2.942) | <0.001 | 1.427 (1.202–1.694) | <0.001 | 1.428 (1.257–1.622) | <0.001 | 1.439 (1.267–1.634) | <0.001 |
| **Mode 2** | 1.679 (1.486–1.896) | <0.001 | 1.410 (1.186–1.675) | <0.001 | 1.463 (1.281–1.671) | <0.001 | 1.462 (1.280–1.671) | <0.001 |
| **Mode 3** | 1.533 (1.357–1.733) | <0.001 | 1.378 (1.159–1.638) | <0.001 | 1.483 (1.283–1.714) | <0.001 | 1.485 (1.285–1.717) | <0.001 |
| **Mode 4** | 1.497 (1.320–1.697) | <0.001 | 1.391 (1.169–1.654) | <0.001 | 1.512 (1.310–1.744) | <0.001 | 1.511 (1.310–1.744) | <0.001 |

Model 1. Unadjusted.

Model 2. Adjusted for demographic data, comorbidities and surgical conditions.

Model 3. Adjusted for variables in model 2 and critical illness severity/management.

Model 4. Adjusted for variables in model 3 and laboratory data.

Abbreviations: PSM, propensity score matching; IPTW, inverse probability of treatment weighting; CBPS, covariate balancing propensity score; HR, hazard ratio; CI, confidence interval.

### Sensitivity analyses using distinct cutoff values for low ALC

We further used distinct cutoff values to define the low ALC in both original and propensity score-matched populations (Fig 3). We found that the association between low ALC and increased mortality existed in a dose-response manner in both the original and propensity score-matched populations.

## Discussion

This study used both Cox regression and the propensity score-based approach to determine the association between week-one ALC and one-year mortality in critically ill surgical patients. We found that low week-one ALC was an independent risk factor for one-year mortality among 8502 critically ill surgical patients. The aforementioned association tended to be more prominent in younger patients and those with fewer comorbidities and lower severities of critical illness. Moreover, the association was consistent in PSM, IPTW, and CBPS analyses. These findings suggest that low week-one ALC, a ready-to-use biomarker in critical care, may serve as a predictor for long-term mortality in critically ill surgical patients.

The long-term outcome has been a growing research interest in critical care medicine because of the advance of critical care with the steady increase in the number of patients surviving critical illness [17,28]. Nevertheless, patients discharged from the ICU may still suffer from the deterioration of health, with higher healthcare utilisation and post-hospital mortality than those hospitalised for diseases other than critical illness [29]. Doherty Z. *et al*. conducted a study in 23 ICUs, mainly medical ICUs, and found that the survival of ICU patients at one-year post-discharge was approximately 90% compared to the age-matched general population cohort of 98% [30]. Few studies focused on the long-term outcome in critically ill surgical patients. Ou *et al*. conducted a population-based study with 1857 critically ill surgical patients with post-operative sepsis and reported that one-year post-hospital survival rate among critically ill surgical patients with post-operative sepsis was 86.3% (1606/18578) [31]. The

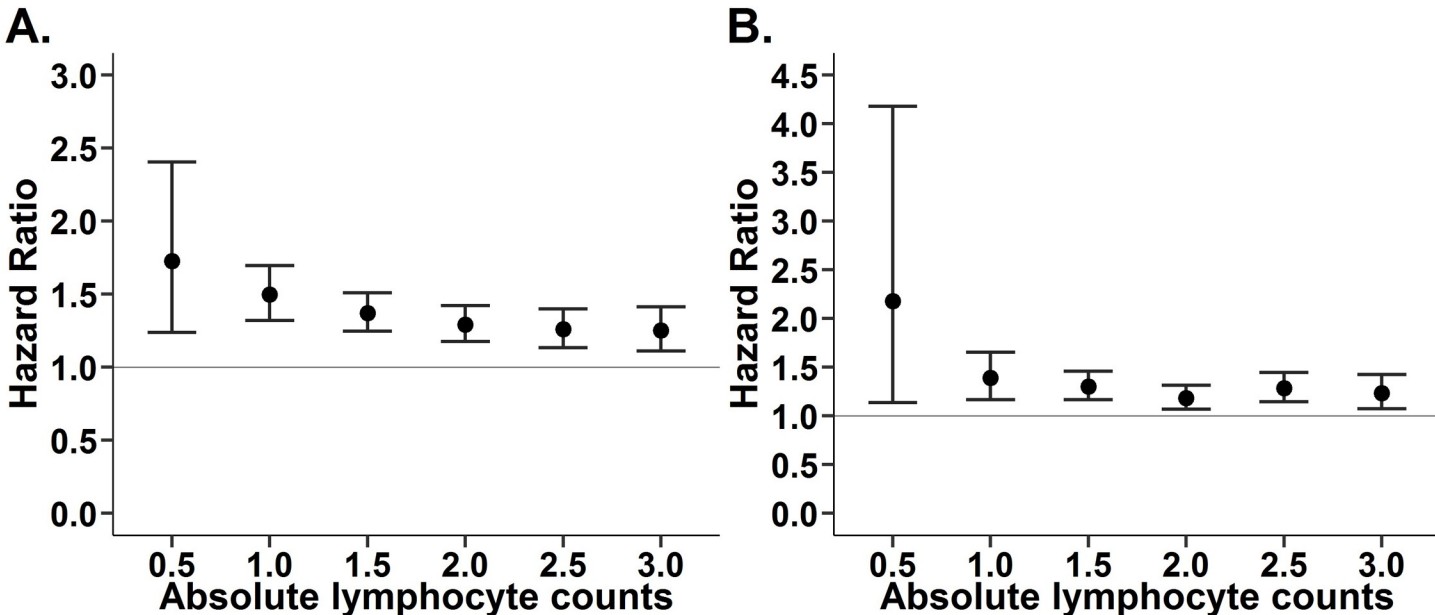

**Fig 3. The strength of association between long-term mortality and week-one absolute lymphocyte count with distinct cutoff values.** (A) Original population (B) Propensity-score matched population.

aforementioned data were similar to our data in that the one-year post-hospital survival rate was 85.6% (6107/7135) among patients discharged from the surgical ICUs in this study (Table 1).

Patients discharged from the ICU may still have persistent inflammation and immunosuppression, including depletion of immune cells, so-called PICS [15,16,32]. Indeed, a number of studies have been conducted to address the association between low ALC/lymphopenia and long-term outcomes in hospitalised patients but not critically ill patients. Andreu-Ballester J.C. et al. analysed data from 58,260 hospital admissions and found that patients with lymphopenia, defined as ALC<1000 counts/μL, during hospital stay had a shorter time to the mortality (67.5 vs 96.9 days, p<0.001) [2]. Similar to our findings in this study, Drewry A.M. et al., analysing 335 patients with bacteremia and sepsis, found that persistent lymphopenia (ALC<1200 counts/μL) on day four after the onset of sepsis was associated with increased 28-day and one-year mortality [7]. One recent study, investigating 5321 patients with coronavirus disease 2019 (COVID-19) infection, also found a lower ALC in deceased patients than those in survived patients (0.9 vs 1.4 k/mm3, p<0.001) [11]. In addition, Warny M. et al. used the Danish cohort with 108,135 participants to demonstrate that incidental lymphopenia, defined as ALC<1100 counts/μL, was associated with a 1.6-fold increase in the risk of mortality [33]. However, these studies focused on hospitalised patients but not critically ill surgical patients. To our knowledge, our study was the first to show a significant association between low week-one ALC and one-year mortality in critically ill surgical patients. With regards to the role of ALC among subgroups of surgical patients, we found that patients of general-colorectal surgery divisions had a slighter higher mortality than those in the cardiovascular surgical division, implicating the potential role of existing infection in patients receiving abdominal surgery. The stratified analyses found no modification effect of surgical divisions on the association between ALC and long-term mortality (Table 3). We further conducted subgroup analysis categorised by surgical types and divisions and found a similar trend among those receiving emergent abdominal surgery, who were critically ill surgical patients with increased risk for existing infection (S3 Table). However, we think that more studies are required to clarify the role of ALC in distinct surgical patients.

Intriguingly, there is no consensus on the cutoff value of lymphocyte counts for lymphopenia. Previous studies have used distinct cutoff values to define a low ALC, ranging from 500 to 1500 counts/μL. We found that the majority of studies exploring the association between low ALC and outcomes have chosen a cutoff value of 1000 counts/μL to define lymphopenia [2,13,14]. Hence, we used the cutoff value of below 1000 counts/μL to define low ALC for our study. Moreover, we used the distinct cutoff values of ALC to show that a low ALC correlated with increased mortality in a dose-response manner (Fig 3).

In the present study, the analysis of the modification effect showed that the impact of low ALC on mortality tended to be stronger in patients with expected good outcomes, including younger patients and those with fewer comorbidities and lower severities of critical illness (Table 3). We think that the aforementioned finding should be interpreted with caution and postulate that the aforementioned finding suggests that low ALC might possibly reflect the lasting compromised immunity with prolonged impact on the recovery from critical illness in patients with lower severity/comorbidity, whereas mortality among those with high severity/comorbidities is mainly determined by the initial critical illness. Therefore, the mortality impact of early ALC tended to be prominent among those with expected good outcome, and more studies are required to validate this finding. Furthermore, we acknowledge that it is somehow difficult to differentiate the role of ALC in short-term outcome from those in long-term outcome, given that low ALC-relevant effects should be lasting.

The low ALC in critically ill patients may be attributed to several biological mechanisms, including increased apoptosis of lymphocytes resulting from critical illness. Apoptosis is the principal mechanism of lymphocyte death in surgical patients and patients with other acute illnesses, leading to lymphocyte loss and low ALC [5,34]. The intense but dysregulated modulation of gene expression, particularly *Bcl*-2 superfamily genes, was found in patients with sepsis [32,35]. Previous studies have shown that pro-apoptotic genes *Bim*, *Bak*, *Bid*, and *Fas* were upregulated, whereas anti-apoptotic genes like *Bcl*-2 and *Bcl-xl* were downregulated in critically ill patients [35,36]. Increasing human and animal studies have demonstrated lymphopenia and increased apoptosis in those receiving surgery or anaesthesia [37,38]. Hotchkiss RS *et al.* compared intestinal tissues obtained from 10 patients with acute traumatic injury to 6 control patients undergoing selective bowel resections and found extensive epithelial damage and lymphocyte apoptosis in the traumatic patients, and the degrees of tissue lymphocyte apoptosis was associated with a decreased circulating lymphocyte count [37]. Yamada *et al.* studied dogs receiving anaesthesia with and without laparotomy and found that lymphopenia was observed in both groups after anaesthesia, with more apoptotic cells in dogs receiving laparotomy than those without laparotomy [38]. These data showed the plausbile mechanisms underlying lymphopenia in surgical patients, and our data further provided clinical evidence of the long-term impact of lymphopenia in critically ill patients.

Unlike Randomized Controlled Trials (RCTs), real-world data, such as the present study, are subject to potential confounders despite the advantage of a high number of patients [39]. Therefore, we have used not only PS-matching but also PS-weighting methods, including IPTW and CBPS, to mitigate the potential confounding effect. PSM has been increasingly applied in the past two decades to mitigate confounding effects on research outcomes; however, selecting a subset of individuals from case and control groups based on similar propensity scores raises concerns about the generalizability of the findings [40]. IPTW addresses this issue by using the inverse of the propensity score as a weighting factor for each participant without excluding anyone, thus reducing concerns about generalizability [41]. Nonetheless, IPTW can still be problematic due to the impact of individuals with extreme weights, and CBPS techniques, which include propensity scores as covariates, are less likely to be influenced by individuals with non-overlapping propensity scores [42]. The overall number of subjects in this study is high, and the standardised mean differences of variables after matching between the two groups with their ALC higher and lower than 1000 counts/μL were less than 0.04 (S1 Table and Fig 1). Therefore, we demonstrated a consistent strength of association between low ALC and long-term mortality among the three aforementioned propensity score-based analyses.

There are limitations in this study. First, the observational nature of the study hindered the ability to establish the causal effect. Therefore, we claim the association, instead of causality, in the present study. Second, the single-centre study and the generalizability of the findings might be a concern. Nevertheless, we enrolled a large number of subjects and employed not only Cox regression but also PSM, IPTW, and CBPS to demonstrate the consistent association between early ALC and long-term mortality in critically ill surgical patients. Third, we could not assess the long-term outcomes other than mortality, such as functional status and quality of life. Future studies should aim to incorporate a broader range of outcomes to provide a more holistic view of patient recovery from critical illness. Fourth, we used the average level of ALC in this study, and the findings were consistent using the first ALC (S2 Table). More studies are warranted to determine the effect of dynamic change of ALC on long-term mortality in critically ill surgical patients.

In conclusion, the consistent association between low ALC levels and increased one-year mortality, as established through methodologies like Cox regression and propensity score

analyses, underscores the crucial role of ALC in critically ill surgical patients. The identification of week-one ALC as an early determinant of long-term outcomes highlights the potential use of week-one ALC as a valuable tool for risk stratification, which may enable more personalized, timely interventions in critically ill surgical patients. Further research is necessary to confirm our findings and to elucidate the biological mechanisms, thereby improving patient management strategies in critical care.

## Supporting information

**S1 Fig. Standardised mean differences of variables in different patient populations.** Abbreviations: PSM, propensity score matching; IPTW, inverse probability of treatment weighting; CBPS, covariate balancing propensity score; APACHE, acute physiology and chronic health evaluation; CCI, Charlson comorbidity index; RRT, renal replacement therapy.
(PDF)

**S2 Fig. Kaplan-Meier survival curves of critically ill surgical patients whose absolute lymphocyte count lower and higher/equal than 1.0 ($10^3$/μL)** (A) Original population (B) Propensity-score matched population.
(PDF)

**S1 Table. Characteristics between the patients categorised by week-1 ALC in the primary cohort and propensity score matched cohort.**
(PDF)

**S2 Table. Cox proportional hazard regression analysis for the association between the first absolute lymphocyte count and long-term mortality in the critically ill surgical patients.**
(PDF)

**S3 Table. Subgroup analysis categorized by surgical types and divisions for the association between absolute lymphocyte count and long-term overall mortality in critically ill surgical patients.**
(PDF)

**S1 Dataset.**
(CSV)

## Acknowledgments

We thank the TCVGH artificial intelligence studio to provide technical help for the TCVGH critical care database.

## Author Contributions

**Conceptualization:** Duc Trieu Ho, The Thach Pham, Chieh-Liang Wu, Ming-Cheng Chan, Wen-Cheng Chao.

**Data curation:** Li-Ting Wong, Chieh-Liang Wu, Ming-Cheng Chan, Wen-Cheng Chao.

**Formal analysis:** Li-Ting Wong, Wen-Cheng Chao.

**Funding acquisition:** Chieh-Liang Wu, Wen-Cheng Chao.

**Investigation:** Li-Ting Wong, Chieh-Liang Wu, Ming-Cheng Chan, Wen-Cheng Chao.

**Methodology:** Li-Ting Wong, Chieh-Liang Wu, Ming-Cheng Chan, Wen-Cheng Chao.

**Project administration:** Chieh-Liang Wu, Ming-Cheng Chan, Wen-Cheng Chao.

**Resources:** Wen-Cheng Chao.

**Writing – original draft:** Duc Trieu Ho, The Thach Pham, Wen-Cheng Chao.

**Writing – review & editing:** Duc Trieu Ho, The Thach Pham, Wen-Cheng Chao.

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
