## [Decision Letter · Decision Letter 0]

14 Nov 2023

PONE-D-23-22982Early absolute lymphocyte count was associated with one-year mortality in critically ill surgical patients: a propensity score-matching and weighting studyPLOS ONE

Dear Dr. Chao,

Thank you for submitting your manuscript to PLOS ONE. After careful consideration, we feel that it has merit but does not fully meet PLOS ONE’s publication criteria as it currently stands. Therefore, we invite you to submit a revised version of the manuscript that addresses the points raised during the review process.

The manuscript has been evaluated by three reviewers, and their comments are available below.

The reviewers have raised a number of concerns that need attention. They request additional information on study design and revisions to the statistical analyses.

Could you please revise the manuscript to carefully address the concerns raised?

We look forward to receiving your revised manuscript.

Kind regards,

Jennifer Tucker, PhD

Staff Editor

PLOS ONE

Journal Requirements:

"No"

Reviewers' comments:

Reviewer's Responses to Questions

**Comments to the Author**

1. Is the manuscript technically sound, and do the data support the conclusions?

Reviewer #1: Yes

Reviewer #2: Partly

Reviewer #3: Yes

2. Has the statistical analysis been performed appropriately and rigorously? 

Reviewer #1: Yes

Reviewer #2: Yes

Reviewer #3: Yes

3. Have the authors made all data underlying the findings in their manuscript fully available?

Reviewer #1: Yes

Reviewer #2: Yes

Reviewer #3: No

4. Is the manuscript presented in an intelligible fashion and written in standard English?

Reviewer #1: Yes

Reviewer #2: Yes

Reviewer #3: Yes

5. Review Comments to the Author

Reviewer #1: Logic of the study could be expressed better. Hemogram derived inflammatory markers are associated with mortality of intensive care population (Journal of Intensive Care Medicine, 2019, 34.6: 511-513.). On the other hand, lymphocyte or lymphocyte based markers are also associated with mortality (Cirugía y cirujanos, 2022, 90.5: 596-601.). Thus, absolute lymphocyte count could be reasonable to study. p values of the hazard ratios along with 95% CI could be expressed in results.

Reviewer #2: The manuscript Early absolute lymphocyte count was associated with one-year mortality in critically ill

surgical patients: a propensity score-matching and weighting study studied the association between lymphocyte count and mortality of the critically ill surgical patients. There are some points to correct.

Unit of Body mass index must be provided in table 1.

Similarly, unit of sex must be stated as n(%). Same for other categorical variables.

Discussion should be improved by discussing lymphocyte or lymphocye including indexs' role in patients requierd intensive care.

Reviewer #3: Review

Comments:

I read with great interest the manuscript “Early absolute lymphocyte count was associated with one-year mortality in critically ill surgical patients: a propensity score-matching and weighting study”. In this manuscript the authors claim an association between absolute lymphopenia during the first week of ICU admission and long-term prognosis. This data is interesting, although it has been shown for other ICU populations and the hypothesis this would not be true for surgical patients and the reason why this research is relevant should be better explained in the background.

Major comments

1. The use of the average value of ALC during the first week although pragmatic might introduce some bias in data interpretation. It is likely that during this first week a shift in lymphocyte counts occur. For the ones where more than 1 data point is available what is the variability of the values? If you just use the first ALC or the shift in ALC do you still have the same results?

2. Although this is an important population of surgical population, it is expected to have very different behaviors for the ones with elective surgery versus urgent surgery. This study has a disproportionate high rate of elective surgeries compared to emergent surgeries. This should be interpreted carefully also in the discussion.

3. In addition, it is likely that for the ones with urgent colo-rectal surgery, sepsis would be a frequent underlying diagnosis. Do you have data on this? Can sepsis explain partially your results?

4. Although interesting, it is unexpected that you found a higher association of low lymphocyte counts with a younger age. Do you have a bias towards age for other variables, such as distribution of type of surgery or sepsis? Did you check for interactions between these relevant variables that might explain this finding? I understand your reasoning in the discussion (line 199-206) to explain this finding, but have you also considered the possibility of a relevant bias here? If what you claim is true have you analyzed the association between ALC (first week, and probably and frequently first days) and short-term mortality?

Minor comments

1. In flowchart 1 maybe the term mortality is not properly applied as you are mentioning absolute events. Death?

2. In the results section (lines 119-126) it would be useful to refer what you consider the non-surviving group (hospital? 90 days? 1-year?) and be clear about that.

6. PLOS authors have the option to publish the peer review history of their article (what does this mean?). If published, this will include your full peer review and any attached files.

Reviewer #1: **Yes: **Ibrahim Karagoz

Reviewer #2: **Yes: **Bahri Ozer

Reviewer #3: No

---

## [Author Response · Author response to Decision Letter 0]

23 Nov 2023

Reviewer #1

Q1. Logic of the study could be expressed better. Hemogram derived inflammatory markers are associated with mortality of intensive care population (Journal of Intensive Care Medicine, 2019, 34.6: 511-513.). On the other hand, lymphocyte or lymphocyte based markers are also associated with mortality (Cirugía y cirujanos, 2022, 90.5: 596-601.). Thus, absolute lymphocyte count could be reasonable to study. 

Reply 

-We thank the reviewer for the thorough reading and the insightful suggestions. We have included the references [Reference 6 and 11] and revised the background as well as the discussion. Please refers to line 42-48 (background) and line 187-189 (discussion). 

-Line 42-48

-A low absolute lymphocyte count (ALC) is one of the immune-compromised biomarkers in critically ill patients and has been implicated with vulnerability for secondary infection or infection resulting from opportunistic pathogens in patients who were admitted to the intensive care unit (ICU) [3-6]. Increasing studies have shown the association between ALC and outcomes in critically ill patients, but the majority of studies were conducted in medical or mixed ICUs [7-11]. Few studies have explored the association between low ALC and outcomes in critically ill surgical patients [12-14]. 

-Line 187-189

One recent study, investigating 5321 patients with coronavirus disease 2019 (COVID-19) infection, also found a lower ALC in deceased patients than those in survived patients (0.9 vs 1.4 k/mm3, p<0.001) [11].

Q2. p values of the hazard ratios along with 95% CI could be expressed in results.

Reply 

-We have added the p values of the HR along with 95% CI in the results. Please refers to line 132-134 and line 151-153. 

-Line 132-134

We found that a low ALC was independently associated with increased one-year mortality (adjHR 1.140, 95%CI 1.091–1.192, p<0.001) after adjusting for covariates (Table 2).

-Line 151-153

The adjusted HRs in original, PSM, IPTW, and CBPS populations were 1.497 (95% CI 1.320–1.697, p<0.001), 1.391 (95% CI 1.169–1.654, p<0.001), 1.512 (95% CI 1.310–1.744, p<0.001), 1.511 (95% CI 1.310–1.744, p<0.001), respectively.

 

Reviewer #2: 

The manuscript Early absolute lymphocyte count was associated with one-year mortality in critically ill surgical patients: a propensity score-matching and weighting study studied the association between lymphocyte count and mortality of the critically ill surgical patients. There are some points to correct.

Q1. Unit of Body mass index must be provided in table 1. 

Reply 

-We thank the reviewer for reminding us of providing the unit of body mass index. We have added the unit (kg/m2) of body mass index in the Table 1. 

Q2. Similarly, unit of sex must be stated as n(%). Same for other categorical variables.

Reply 

-We have presented the categorical variables as N (%). 

Q3. Discussion should be improved by discussing lymphocyte or lymphocye including indexs' role in patients required intensive care.

Reply 

-We have added the discussion with regard to the potential rationale (Line 229-239) and clinical relevance (Line 165-167) of ALC in critically ill patients. 

-Line 229-239 (potential rationales)

-Increasing human and animal studies have demonstrated lymphopenia and increased apoptosis in those receiving surgery or anaesthesia [30, 31]. Hotchkiss RS et al. compared intestinal tissues obtained from 10 patients with acute traumatic injury to 6 control patients undergoing selective bowel resections and found extensive epithelial damage and lymphocyte apoptosis in the traumatic patients, and the degrees of tissue lymphocyte apoptosis was associated with a decreased circulating lymphocyte count [30]. Yamada et al. studied dogs receiving anaesthesia with and without laparotomy and found that lymphopenia was observed in both groups after anaesthesia, with more apoptotic cells in dogs receiving laparotomy than those without laparotomy [31]. These data showed the plausbile mechanisms underlying lymphopenia in surgical patients, and our data further provided clinical evidence of the long-term impact of lymphopenia in critically ill patients. 

-Line 165-167 (clinical relevance)

These findings suggest that low week-one ALC, a ready-to-use biomarker in critical care, may serve as a predictor for long-term mortality in critically ill surgical patients. 

 

Reviewer #3: 

Comments:

I read with great interest the manuscript “Early absolute lymphocyte count was associated with one-year mortality in critically ill surgical patients: a propensity score-matching and weighting study”. In this manuscript the authors claim an association between absolute lymphopenia during the first week of ICU admission and long-term prognosis. This data is interesting, although it has been shown for other ICU populations and the hypothesis this would not be true for surgical patients and the reason why this research is relevant should be better explained in the background.

Major comments

Q1. The use of the average value of ALC during the first week although pragmatic might introduce some bias in data interpretation. It is likely that during this first week a shift in lymphocyte counts occur. For the ones where more than 1 data point is available what is the variability of the values? If you just use the first ALC or the shift in ALC do you still have the same results?

Reply 

-We thank the reviewer for this comment and have conducted the analysis using the first level of ALC, and the data were consistent with the analysis using the average during the first week (Supplemental Table 2). Additionally, we have acknowledged that we could not detailly clarify the dynamic change of ALC in this study. Please refer to the supplemental Table 2 and the limitation section (Line 261-264).

-Line 261-264 (limitation section) 

Fourth, we used the average level of ALC in this study, and the data were consistent using the first ALC (Supplemental Table 2). More studies are warranted to explore the impact of dynamic change of ALC on long-term mortality in critically ill surgical patients. 

Q2. Although this is an important population of surgical population, it is expected to have very different behaviors for the ones with elective surgery versus urgent surgery. This study has a disproportionate high rate of elective surgeries compared to emergent surgeries. This should be interpreted carefully also in the discussion.

Reply 

-We totally agree with the reviewer that we should interpret this point with caution. We have conducted additional analysis subgroup analysis categorised by surgical types and divisions and found a similar trend among those receiving emergent abdominal surgery (please refers to Q3). We have added a discussion and hedge statement with regard to this crucial point; please refers to line 194-202. Given that we have matched surgical divisions and types of surgery in the propensity score-based analysis, the main claim of the association between ALC and long-term mortality in the study population should not be affected by the surgical division/type.

-Line 194-202 

With regards to the role of ALC among subgroups of surgical patients, we found that patients of general-colorectal surgery divisions had a slighter higher mortality than those in the cardiovascular surgical division, implicating the potential role of existing infection in patients receiving abdominal surgery. The stratified analyses found no modification effect of surgical divisions on the association between ALC and long-term mortality (Table 3). We further conducted subgroup analysis categorised by surgical types and divisions and found a similar trend among those receiving emergent abdominal surgery, who were critically ill surgical patients with increased risk for existing infection (Supplemental Table 3). However, we think that more studies are required to clarify the role of ALC in distinct surgical patients 

Q3. In addition, it is likely that for the ones with urgent colo-rectal surgery, sepsis would be a frequent underlying diagnosis. Do you have data on this? Can sepsis explain partially your results?

Reply 

-We must thank the reviewer for pointing out this point, and the analysis does show the consistent association between ALC and long-term mortality in patients receiving emergent colo-rectal surgery. To avoid redundancy, please refers to the supplemental Table 3 and the reply to Q2. 

Q4. Although interesting, it is unexpected that you found a higher association of low lymphocyte counts with a younger age. Do you have a bias towards age for other variables, such as distribution of type of surgery or sepsis? Did you check for interactions between these relevant variables that might explain this finding? I understand your reasoning in the discussion (line 199-206) to explain this finding, but have you also considered the possibility of a relevant bias here? If what you claim is true have you analysed the association between ALC (first week, and probably and frequently first days) and short-term mortality?

Reply 

-We thank the reviewer for this comment on the result regarding the effect of modification, which was also our concern in the manuscript preparation. The focus of this study is the association between ALC and long-term mortality in critically ill surgical patients, and we have used both Cox regression and propensity-based analyses to demonstrate the association between ALC and long-term mortality. The effect of modification is the additional analysis based on the Cox regression, and we totally agree with the reviewer that we should interpret the finding with caution. We have revised the discussion and added a number of hedge statements regarding this point. Please refers to line 210-221 for the revised discussion with hedge statements. But, this study not only used Cox regression but also applied the propensity score-based analyses to demonstrate the consistent association between ALC and long-term mortality in critically ill surgical patients. 

-Line 210-221

In the present study, the analysis of the modification effect showed that the impact of low ALC on mortality tended to be stronger in patients with expected good outcomes, including younger patients and those with fewer comorbidities and lower severities of critical illness (Table 3). We think that the aforementioned finding should be interpreted with caution and postulate that the aforementioned finding suggests that low ALC might possibly reflect the lasting compromised immunity with prolonged impact on the recovery from critical illness in patients with lower severity/comorbidity, whereas mortality among those with high severity/comorbidities is mainly determined by the initial critical illness. Therefore, the mortality impact of early ALC tended to be prominent among those with expected good outcome, and more studies are required to validate this finding. Furthermore, we acknowledge that it is somehow difficult to differentiate the role of ALC in short-term outcome from those in long-term outcome, given that low ALC-relevant effects should be lasting.

Minor comments

Q5. In flowchart 1 maybe the term mortality is not properly applied as you are ment oning absolute events. Death?

Reply 

-We thank the reviewer for this suggestion for explicit wording. We have replaced the “mortality” with “death” in the Fig. 1. 

Q6. In the results section (lines 119-126) it would be useful to refer what you consider the non-surviving group (hospital? 90 days? 1-year?) and be clear about that.

Reply 

-We thank the reviewer for this suggestion to avoid ambiguity and have revised the sentence by replacing “non-surviving group” with “patients deceased within one year”. (please refers to line 119)

---

## [Decision Letter · Decision Letter 1]

9 Apr 2024

PONE-D-23-22982R1Early absolute lymphocyte count was associated with one-year mortality in critically ill surgical patients: a propensity score-matching and weighting studyPLOS ONE

Dear Dr. Chao,

Thank you for submitting your manuscript to PLOS ONE. After careful consideration, we feel that it has merit but does not fully meet PLOS ONE’s publication criteria as it currently stands. Therefore, we invite you to submit a revised version of the manuscript that addresses the points raised during the review process. Please submit your revised manuscript by May 24 2024 11:59PM. If you will need more time than this to complete your revisions, please reply to this message or contact the journal office at plosone@plos.org. Please include the following items when submitting your revised manuscript:A rebuttal letter that responds to each point raised by the academic editor and reviewer(s). You should upload this letter as a separate file labeled 'Response to Reviewers'.A marked-up copy of your manuscript that highlights changes made to the original version. You should upload this as a separate file labeled 'Revised Manuscript with Track Changes'.An unmarked version of your revised paper without tracked changes. You should upload this as a separate file labeled 'Manuscript'.If applicable, we recommend that you deposit your laboratory protocols in protocols.io to enhance the reproducibility of your results. Protocols.io assigns your protocol its own identifier (DOI) so that it can be cited independently in the future. For instructions see: https://journals.plos.org/plosone/s/submission-guidelines#loc-laboratory-protocols. Additionally, PLOS ONE offers an option for publishing peer-reviewed Lab Protocol articles, which describe protocols hosted on protocols.io. Read more information on sharing protocols at https://plos.org/protocols?utm_medium=editorial-email&utm_source=authorletters&utm_campaign=protocols.

We look forward to receiving your revised manuscript.

Kind regards,

*
**Ali Amanati**
*

Academic Editor

*
**PLOS ONE**
*

Journal Requirements:

Additional Editor Comments:

Dear authors,

‎

New comments were posted by the reviewer #4 and reviewer #5. So, the manuscripts require a ‎round of revision.‎ Please provide a point-by-point response to the reviewers' ‎comments and highlight all the ‎amends on your manuscript with yellow color.‎

Yours

Reviewers' comments:

Reviewer's Responses to Questions

**Comments to the Author**

1. If the authors have adequately addressed your comments raised in a previous round of review and you feel that this manuscript is now acceptable for publication, you may indicate that here to bypass the “Comments to the Author” section, enter your conflict of interest statement in the “Confidential to Editor” section, and submit your "Accept" recommendation.

Reviewer #1: All comments have been addressed

Reviewer #2: All comments have been addressed

Reviewer #4: All comments have been addressed

Reviewer #5: All comments have been addressed

Reviewer #6: All comments have been addressed

2. Is the manuscript technically sound, and do the data support the conclusions?

Reviewer #1: Yes

Reviewer #2: Yes

Reviewer #4: Yes

Reviewer #5: Partly

Reviewer #6: Yes

3. Has the statistical analysis been performed appropriately and rigorously? 

Reviewer #1: Yes

Reviewer #2: Yes

Reviewer #4: Yes

Reviewer #5: Yes

Reviewer #6: Yes

4. Have the authors made all data underlying the findings in their manuscript fully available?

Reviewer #1: Yes

Reviewer #2: Yes

Reviewer #4: Yes

Reviewer #5: Yes

Reviewer #6: Yes

5. Is the manuscript presented in an intelligible fashion and written in standard English?

Reviewer #1: Yes

Reviewer #2: Yes

Reviewer #4: Yes

Reviewer #5: Yes

Reviewer #6: Yes

6. Review Comments to the Author

Reviewer #1: PONE-D-23-22982R1 id no article is carefully re-evaluated. Authors' revisions are satisfactory. It is publishable in the journal in the latest form.

Reviewer #2: Authors adequately revised the manuscript according to my review recommendations. No other points to suggest further revision. I think it is acceptable for publication.

Reviewer #4: Overall, the paper appears well-structured and comprehensive, covering various aspects of the study from methods to results and discussion. Here are some suggestions for improvement:

1- Ensure that the language throughout the paper is clear and precise. For example, in the Methods section, you could clarify the term "week-one ALC" to ensure readers understand it refers to the absolute lymphocyte count within the first week of ICU admission.

2- The statement regarding ethical approval is clear, but you could briefly mention any specific ethical considerations or guidelines followed during the study, especially concerning patient data and privacy.

3- It's important to provide details about how covariates were selected and why they were considered relevant. This could enhance the understanding of the readers regarding the adjustments made in the analysis.

4- While you've described the statistical methods used, consider providing a bit more context on why each method was chosen and how it contributes to the analysis.

5- Results Interpretation: Ensure that the results are interpreted accurately and avoid overgeneralization. For instance, when discussing mortality rates, it may be helpful to provide context by comparing them to similar studies or national averages.

6- Discussion: The discussion is insightful, but you could expand on potential implications of the findings for clinical practice or future research directions. Additionally, consider discussing any limitations in more detail and how they may have influenced the results.

7- Conclusion: The conclusion summarizes the findings well, but you could reiterate the key implications of the study and emphasize the need for further research to validate the findings and explore underlying mechanisms.

Reviewer #5: The manuscript is highly valuable and well-written. The reviewers' comments were correctly and fairly fully applied, except for a few that have received little attention. It is anticipated that applying these suggestions will enhance the article.

Reviewer #3:

In this manuscript, the authors claim an association

between absolute lymphopenia during the first week of ICU admission and long-term

prognosis. This data is interesting, although it has been shown for other ICU populations and the hypothesis this would not be true for surgical patients and the reason why this research is relevant should be better explained in the background.

It is preferable to more explain the importance of the topic for surgical patients in the background section.

Q6. In the results section (lines 119-126) it would be useful to refer what you consider the non-surviving group (hospital? 90 days? 1-year?) and be clear about that.

The phrase "patients deceased within one year" should not be replaced with the non-survivor group, but rather should be included in the explanation since the term non-survivor group is referenced later.

Two supplementary remarks are provided below, which the authors should take into consideration:

1- In all related tables, the term "surgical patients" should be mentioned, for example, table 2 should be modified.

2- In the title, the term "absolute lymphocyte " is referenced, whereas, in the background, the study's objective is delineated as investigating the association between week-one ALC and long-term mortality, which appears to need correction.

Reviewer #6: The article is well-written and structured, providing an in-depth analysis of the association between week-one ALC and one-year mortality in critically ill surgical patients. The authors have used appropriate statistical methodology to evaluate the desired association and have performed sensitivity analyses with different analytical strategies. Moreover, the authors have addressed all the concerns of the reviewers properly and comprehensively.

7. PLOS authors have the option to publish the peer review history of their article (what does this mean?). If published, this will include your full peer review and any attached files.

Reviewer #1: No

Reviewer #2: No

Reviewer #4: No

Reviewer #5: No

Reviewer #6: No

---

## [Author Response · Author response to Decision Letter 1]

16 Apr 2024

Reviewers' comment

# Reviewer #1: 

PONE-D-23-22982R1 id no article is carefully re-evaluated. Authors' revisions are satisfactory. It is publishable in the journal in the latest form.

Reply 

- We thank the reviewer for the thorough reading and the insightful suggestions in the process of revision.

# Reviewer #2: 

Authors adequately revised the manuscript according to my review recommendations. No other points to suggest further revision. I think it is acceptable for publication.

Reply 

- We thank the reviewer for the thorough reading and the insightful suggestions in the process of revision.

# Reviewer #3:

In this manuscript, the authors claim an association between absolute lymphopenia during the first week of ICU admission and long-term prognosis. This data is interesting, although it has been shown for other ICU populations and the hypothesis this would not be true for surgical patients and the reason why this research is relevant should be better explained in the background.

Q1. It is preferable to more explain the importance of the topic for surgical patients in the background section.

Reply 

- We thank the reviewer for this suggestion that we should elaborate on the research niche, lack of evidence focusing on long-term outcome in critically ill surgical patients, and we have revised the background. Please refers to line 45-58

- Line 45-58

- Increasing studies have shown the association between ALC and outcomes in critically ill patients, but the majority of studies were conducted in medical or mixed ICUs [7-11]. Few studies have explored the association between low ALC and outcomes in critically ill surgical patients [12-14]. 

- Notably, the aforementioned low ALC-associated immunosuppressed state in acute illness may last after the stabilization of acute illness and contribute to prolonged pathophysiological abnormalities among patients discharged from the ICUs, which have been known as post-intensive care syndrome (PICS) [15, 16]. Additionally, significant progress in critical care medicine over the past twenty years has not only decreased mortality rates in ICUs but also resulted in a steadily growing number of patients being discharged from these units [17]. Increasing studies, including our previous studies, have addressed the long-term outcomes of critically ill patients in medical ICUs [18-20]. However, the role of early low ALC in the long-term outcome among critically ill patients, particularly those who were admitted to surgical ICUs, remains unclear. 

Q2. In the results section (lines 119-126) it would be useful to refer what you consider the non-surviving group (hospital? 90 days? 1-year?) and be clear about that. The phrase "patients deceased within one year" should not be replaced with the non-survivor group, but rather should be included in the explanation since the term non-survivor group is referenced later.

Reply 

- We thank the reviewer that we should precisely describe the data, particularly in this study focused on the long-term outcome in critically ill patients. To avoid misunderstanding of the “non-surviving group”, we have replaced “non-surviving group” with “patients deceased within one year” as suggested. Please refers to line 127, 130, and 135-316. 

- 

Q3. Two supplementary remarks are provided below, which the authors should take into consideration:

Q3-1. In all related tables, the term "surgical patients" should be mentioned, for example, table 2 should be modified.

Reply 

- We thank the reviewer for this suggestion and have revised the manuscript as suggested. Please refers to the revised table 2, supplemental table 3, and the legend of supplemental figure 2. 

Q3-2. In the title, the term "absolute lymphocyte count" is referenced, whereas, in the background, the study's objective is delineated as investigating the association between week-one ALC and long-term mortality, which appears to need correction.

Reply 

- We thank the reviewer for this suggestion and have revised the manuscript as suggested. Please refers to line 59.

# Reviewer #4: 

Overall, the paper appears well-structured and comprehensive, covering various aspects of the study from methods to results and discussion. Here are some suggestions for improvement:

Q1- Ensure that the language throughout the paper is clear and precise. For example, in the Methods section, you could clarify the term "week-one ALC" to ensure readers understand it refers to the absolute lymphocyte count within the first week of ICU admission.

Reply 

- We thank the reviewer-4 and reviewer-3 that we should precisely describe the exposure in this study. We have revised the manuscript as suggested. Please refers to line 74, 77, and 78. 

Q2- The statement regarding ethical approval is clear, but you could briefly mention any specific ethical considerations or guidelines followed during the study, especially concerning patient data and privacy.

Reply 

- We have added that this study was performed in accordance with the Declaration of Helsinki. Please refers to line 66-68. 

- Line 66-68

The study was performed in accordance with the Declaration of Helsinki [21]. The Institutional Review Board of the TCVGH approved the study and waived the informed consent because all of the data were de-identified (Approval No. CE23045C). 

Q3- It's important to provide details about how covariates were selected and why they were considered relevant. This could enhance the understanding of the readers regarding the adjustments made in the analysis.

Reply 

- We thank the reviewer for this suggestion and have added details on the selection of variables in the Cox regression. Please refers to line 96-99. 

- Line 96-99

Variables were included in the multivariable model if the associated univariable p value was < 0.20 and the variance inflation factor was < 10 [23]. We further included white blood cell count, given that white blood cell count appears to be a known prognosis-relevant variable in critically ill patients [24]

Q4- While you've described the statistical methods used, consider providing a bit more context on why each method was chosen and how it contributes to the analysis.

Reply 

- We are grateful for this suggestion to highlight that we employed both propensity score-matching and -weighting methods in this study and found a robust association between week-one absolute lymphocyte count and one-year mortality in critically ill surgical patients. The revised details with regard to the statistical methods are now added in line 102-106 (methodology) and line 254-262 (discussion section). 

- Line 102-106

We further employed not only propensity score-matching (PSM) but also propensity score-weighting methods, including the inverse probability of treatment weighting (IPTW) and the covariate balancing propensity score (CBPS), to confirm the relationship between the low ALC and high one-year mortality among the enrolled critically ill surgical patients [25-27].

- Line 254-262

PSM has been increasingly applied in the past two decades to mitigate confounding effects on research outcomes; however, selecting a subset of individuals from case and control groups based on similar propensity scores raises concerns about the generalizability of the findings [40]. IPTW addresses this issue by using the inverse of the propensity score as a weighting factor for each participant without excluding anyone, thus reducing concerns about generalizability [41]. Nonetheless, IPTW can still be problematic due to the impact of individuals with extreme weights, and CBPS techniques, which include propensity scores as covariates, are less likely to be influenced by individuals with non-overlapping propensity scores [42].

Q5- Results Interpretation: Ensure that the results are interpreted accurately and avoid overgeneralization. For instance, when discussing mortality rates, it may be helpful to provide context by comparing them to similar studies or national averages.

Reply 

- We thank the reviewer for pointing out that we should further elaborate the generalization issue of our findings. The revised discussion on this issue has been added in line 183-188.

- Line 183-188

Few studies focused on the long-term outcome in critically ill surgical patients. Ou et al. conducted a population-based study with 1857 critically ill surgical patients with post-operative sepsis and reported that the one-year post-hospital survival rate among critically ill surgical patients with post-operative sepsis was 86.3% (1606/18578) [31]. The aforementioned data were similar to our data in that the one-year post-hospital survival rate was 85.6% (6107/7135) among patients discharged from the surgical ICUs in this study (Table 1).

Q6- Discussion: The discussion is insightful, but you could expand on potential implications of the findings for clinical practice or future research directions. Additionally, consider discussing any limitations in more detail and how they may have influenced the results.

Reply 

- We have revised the conclusion section, please refers to the reply of Q7. Additionally, we have expanded the discussion regarding the limitations of this study. Please refers to line 272-278 in the limitation section. 

- Line 272-278 

Third, we could not assess the long-term outcomes other than mortality, such as functional status and quality of life. Future studies should aim to incorporate a broader range of outcomes to provide a more holistic view of patient recovery from critical illness. Fourth, we used the average level of ALC in this study, and the findings were consistent using the first ALC (Supplemental Table 2). More studies are warranted to determine the effect of dynamic change of ALC on long-term mortality in critically ill surgical patients.

Q7- Conclusion: The conclusion summarizes the findings well, but you could reiterate the key implications of the study and emphasize the need for further research to validate the findings and explore underlying mechanisms.

Reply 

- We thank the reviewer for the suggestion that we should point out the practical implication and future work in the conclusion section. Please refers to line 279-286

- Line 279-286

In conclusion, the consistent association between low ALC levels and increased one-year mortality, as established through methodologies like Cox regression and propensity score analyses, underscores the crucial role of ALC in critically ill surgical patients. The identification of week-one ALC as an early determinant of long-term outcomes highlights the potential use of week-one ALC as a valuable tool for risk stratification, which may enable more personalized, timely interventions in critically ill surgical patients. Further research is necessary to confirm our findings and to elucidate the biological mechanisms, thereby improving patient management strategies in critical care.

 

# Reviewer #5: The manuscript is highly valuable and well-written. The reviewers' comments were correctly and fairly fully applied, except for a few that have received little attention. It is anticipated that applying these suggestions will enhance the article.

Reply 

- We thank the reviewer for the thorough reading and the insightful suggestions in the revision process.

# Reviewer #6: The article is well-written and structured, providing an in-depth analysis of the association between week-one ALC and one-year mortality in critically ill surgical patients. The authors have used appropriate statistical methodology to evaluate the desired association and have performed sensitivity analyses with different analytical strategies. Moreover, the authors have addressed all the concerns of the reviewers properly and comprehensively.

Reply 

- We thank the reviewer for the thorough reading and the insightful suggestions in the revision process.

---

## [Decision Letter · Decision Letter 2]

15 May 2024

Early absolute lymphocyte count was associated with one-year mortality in critically ill surgical patients: a propensity score-matching and weighting study

PONE-D-23-22982R2

Dear Dr. Wen-Cheng Chao,

We’re pleased to inform you that your manuscript has been judged scientifically suitable for publication and will be formally accepted for publication once it meets all outstanding technical requirements.

Kind regards,

*
**Ali Amanati**
*

Academic Editor

*
**PLOS ONE**
*

Additional Editor Comments (optional):

Editor comments ‎

The current article is scientifically valid in its current form. So, based on my ‎opinion and the respected reviewers' comments could be published.‎

Reviewers' comments:

Reviewer's Responses to Questions

**Comments to the Author**

1. If the authors have adequately addressed your comments raised in a previous round of review and you feel that this manuscript is now acceptable for publication, you may indicate that here to bypass the “Comments to the Author” section, enter your conflict of interest statement in the “Confidential to Editor” section, and submit your "Accept" recommendation.

Reviewer #4: All comments have been addressed

Reviewer #5: All comments have been addressed

2. Is the manuscript technically sound, and do the data support the conclusions?

Reviewer #4: Yes

Reviewer #5: Yes

3. Has the statistical analysis been performed appropriately and rigorously? 

Reviewer #4: Yes

Reviewer #5: Yes

4. Have the authors made all data underlying the findings in their manuscript fully available?

Reviewer #4: Yes

Reviewer #5: Yes

5. Is the manuscript presented in an intelligible fashion and written in standard English?

Reviewer #4: Yes

Reviewer #5: Yes

6. Review Comments to the Author

Reviewer #4: After a comprehensive review, I feel that this manuscript is now acceptable for publication. I appreciate the authors for addressing the existing concerns

Reviewer #5: The authors' revisions are satisfactory. I don't have any other points to suggest for further revision. I believe the manuscript is now acceptable for publication.

7. PLOS authors have the option to publish the peer review history of their article (what does this mean?). If published, this will include your full peer review and any attached files.

Reviewer #4: No

Reviewer #5: No

---

## [Editor Report · Acceptance letter]

20 May 2024

PONE-D-23-22982R2 

PLOS ONE

Dear Dr. Chao, 

I'm pleased to inform you that your manuscript has been deemed suitable for publication in PLOS ONE. Congratulations! Your manuscript is now being handed over to our production team.

Kind regards, 

on behalf of

Professor Ali Amanati 

Academic Editor

PLOS ONE